# Adaptive Skills Adaptive Partitions (ASAP)

**Daniel J. Mankowitz, Timothy A. Mann**[*] **and Shie Mannor**
The Technion - Israel Institute of Technology,
Haifa, Israel
danielm@tx.technion.ac.il, mann.timothy@acm.org, shie@ee.technion.ac.il
[*]Timothy Mann now works at Google Deepmind.

## Abstract

We introduce the Adaptive Skills, Adaptive Partitions (ASAP) framework that (1) learns skills (i.e., temporally extended actions or options) as well as (2) where to apply them. We believe that both (1) and (2) are necessary for a truly general skill learning framework, which is a key building block needed to scale up to lifelong learning agents. The ASAP framework can also solve related new tasks simply by adapting where it applies its existing learned skills. We prove that ASAP converges to a local optimum under natural conditions. Finally, our experimental results, which include a RoboCup domain, demonstrate the ability of ASAP to learn where to reuse skills as well as solve multiple tasks with considerably less experience than solving each task from scratch.

## 1 Introduction

Human-decision making involves decomposing a task into a course of action. The course of action is typically composed of abstract, high-level actions that may execute over different timescales (e.g., walk to the door or make a cup of coffee). The decision-maker chooses actions to execute to solve the task. These actions may need to be *reused* at different points in the task. In addition, the actions may need to be used across multiple, related tasks.

Consider, for example, the task of building a city. The course of action to building a city may involve building the foundations, laying down sewage pipes as well as building houses and shopping malls. Each action operates over multiple timescales and certain actions (such as building a house) may need to be reused if additional units are required. In addition, these actions can be reused if a neighboring city needs to be developed (multi-task scenario).

Reinforcement Learning (RL) represents actions that last for multiple timescales as Temporally Extended Actions (TEAs) (Sutton et al., 1999), also referred to as options, skills (Konidaris & Barto, 2009) or macro-actions (Hauskrecht, 1998). It has been shown both experimentally (Precup & Sutton, 1997; Sutton et al., 1999; Silver & Ciosek, 2012; Mankowitz et al., 2014) and theoretically (Mann & Mannor, 2014) that TEAs speed up the convergence rates of RL planning algorithms. TEAs are seen as a potentially viable solution to making RL truly scalable. TEAs in RL have become popular in many domains including RoboCup soccer (Bai et al., 2012), video games (Mann et al., 2015) and Robotics (Fu et al., 2015). Here, decomposing the domains into temporally extended courses of action (strategies in RoboCup, move combinations in video games and skill controllers in Robotics for example) has generated impressive solutions. From here on in, we will refer to TEAs as skills.

A course of action is defined by a policy. A policy is a solution to a Markov Decision Process (MDP) and is defined as a mapping from states to a probability distribution over actions. That is, it tells the RL agent which action to perform given the agent's current state. We will refer to an *inter*-skill policy as being a policy that tells the agent which skill to execute, given the current state.

A truly general skill learning framework must (1) learn skills as well as (2) automatically compose them together (as stated by Bacon & Precup (2015)) and determine where each skill should be executed (the inter-skill policy). This framework should also determine (3) where skills can be reused

Table 1: Comparison of Approaches to ASAP

| | Automated Skill Learning with Policy Gradient | Automatic Skill Composition | Continuous State Multitask Learning | Learning Reusable Skills | Correcting Model Misspecification |
|---|---|---|---|---|---|
| ASAP (this paper) | ✓ | ✓ | ✓ | ✓ | ✓ |
| da Silva et al. 2012 | ✓ | × | ✓ | × | × |
| Konidaris & Barto 2009 | × | ✓ | × | × | × |
| Bacon & Precup 2015 | ✓ | × | × | × | × |
| Eaton & Ruvolo 2013 | × | × | ✓ | × | × |

in different parts of the state space and (4) adapt to changes in the task itself. Finally it should also be able to (5) correct *model misspecification* (Mankowitz et al., 2014). Whilst different forms of model misspecification exist in RL, we define it here as having an unsatisfactory set of skills and inter-skill policy that provide a sub-optimal solution to a given task. This skill learning framework should be able to correct this misspecification to obtain a near-optimal solution. A number of works have addressed some of these issues separately as shown in Table 1. However, no work, to the best of our knowledge, has combined all of these elements into a truly general skill-learning framework.

Our framework entitled 'Adaptive Skills, Adaptive Partitions (ASAP)' is the first of its kind to incorporate all of the above-mentioned elements into a single framework, as shown in Table 1, and solve continuous state MDPs. It receives as input a misspecified model (a sub-optimal set of skills and *inter*-skill policy). The ASAP framework corrects the misspecification by simultaneously learning a near-optimal skill-set and inter-skill policy which are both stored, in a Bayesian-like manner, within the ASAP policy. In addition, ASAP automatically composes skills together, learns where to reuse them and learns skills across multiple tasks.

**Main Contributions: (1)** The Adaptive Skills, Adaptive Partitions (ASAP) algorithm that automatically corrects a misspecified model. It learns a set of near-optimal skills, automatically composes skills together and learns an inter-skill policy to solve a given task. **(2)** Learning skills over multiple different tasks by automatically adapting both the inter-skill policy and the skill set. **(3)** ASAP can determine where skills should be reused in the state space. **(4)** Theoretical convergence guarantees.

## 2 Background

**Reinforcement Learning Problem:** A Markov Decision Process is defined by a 5-tuple $\langle X, A, R, \gamma, P \rangle$ where $X$ is the state space, $A$ is the action space, $R \in [-b, b]$ is a bounded reward function, $\gamma \in [0, 1]$ is the discount factor and $P : X \times A \to [0, 1]^X$ is the transition probability function for the MDP. The solution to an MDP is a policy $\pi : X \to \Delta_A$ which is a function mapping states to a probability distribution over actions. An optimal policy $\pi^* : X \to \Delta_A$ determines the best actions to take so as to maximize the expected reward. The value function $V^{\pi}(x) = \mathbb{E}_{a \sim \pi(\cdot|a)} \left[ R(x, a) \right] + \gamma E_{x' \sim P(\cdot|x,a)} \left[ V^{\pi}(x') \right]$ defines the expected reward for following a policy $\pi$ from state $x$. The optimal expected reward $V^{\pi^*}(x)$ is the expected value obtained for following the optimal policy from state $x$.

**Policy Gradient:** Policy Gradient (PG) methods have enjoyed success in recent years especially in the fields of robotics (Peters & Schaal, 2006, 2008). The goal in PG is to learn a policy $\pi_\theta$ that maximizes the expected return $J(\pi_\theta) = \int_\tau P(\tau) R(\tau) d\tau$, where $\tau$ is a set of trajectories, $P(\tau)$ is the probability of a trajectory and $R(\tau)$ is the reward obtained for a particular trajectory. $P(\tau)$ is defined as $P(\tau) = P(x_0) \Pi_{k=0}^{T} P(x_{k+1}|x_k, a_k) \pi_\theta(a_k|x_k)$. Here, $x_k \in X$ is the state at the $k^{th}$ timestep of the trajectory; $a_k \in A$ is the action at the $k^{th}$ timestep; $T$ is the trajectory length. Only the policy, in the general formulation of policy gradient, is parameterized with parameters $\theta$. The idea is then to update the policy parameters using stochastic gradient descent leading to the update rule $\theta_{t+1} = \theta_t + \eta \nabla J(\pi_\theta)$, where $\theta_t$ are the policy parameters at timestep $t$, $\nabla J(\pi_\theta)$ is the gradient of the objective function with respect to the parameters and $\eta$ is the step size.

## 3 Skills, Skill Partitions and Intra-Skill Policy

**Skills:** A skill is a parameterized *Temporally Extended Action* (TEA) (Sutton et al., 1999). The power of a skill is that it incorporates both *generalization* (due to the parameterization) and *temporal*

*abstraction.* Skills are a special case of options and therefore inherit many of their useful theoretical properties (Sutton et al., 1999; Precup et al., 1998).

**Definition 1.** *A Skill $\zeta$ is a TEA that consists of the two-tuple $\zeta = \langle \sigma_\theta, p(x) \rangle$ where $\sigma_\theta : X \to \Delta A$ is a parameterized, intra-skill policy with parameters $\theta \in \mathbb{R}^d$ and $p : X \to [0, 1]$ is the termination probability distribution of the skill.*

**Skill Partitions:** A skill, by definition, performs a specialized task on a sub-region of a state space. We refer to these sub-regions as Skill Partitions (SPs) which are necessary for skills to specialize during the learning process. A given set of SPs covering a state space effectively define the inter-skill policy as they determine where each skill should be executed. These partitions are unknown *a-priori* and are generated using intersections of hyperplane half-spaces (described below). Hyperplanes provide a natural way to automatically compose skills together. In addition, once a skill is being executed, the agent needs to select actions from the skill's intra-skill policy $\sigma_\theta$. We next utilize SPs and the intra-skill policy for each skill to construct the ASAP policy, defined in Section 4. We first define a skill hyperplane.

**Definition 2.** *Skill Hyperplane (SH): Let $\psi_{x,m} \in \mathbb{R}^d$ be a vector of features that depend on a state $x \in X$ and an MDP environment $m$. Let $\beta_i \in \mathbb{R}^d$ be a vector of hyperplane parameters. A skill hyperplane is defined as $\psi_{x,m}^T \beta_i = L$, where $L$ is a constant.*

In this work, we interpret hyperplanes to mean that the intersection of *skill* hyperplane half spaces form sub-regions in the state space called Skill Partitions (SPs), defining where each skill is executed. Figure 1a contains two example skill hyperplanes $h_1, h_2$. Skill $\zeta_1$ is executed in the SP defined by the intersection of the positive half-space of $h_1$ and the negative half-space of $h_2$. The same argument applies for $\zeta_0, \zeta_2, \zeta_3$. From here on in, we will refer to skill $\zeta_i$ interchangeably with its index $i$.

Skill hyperplanes have two functions: (1) They automatically compose skills together, creating chainable skills as desired by Bacon & Precup (2015). (2) They define SPs which enable us to derive the probability of executing a skill, given a state $x$ and MDP $m$. First, we need to be able to uniquely identify a skill. We define a binary vector $B = [b_1, b_2, \cdots, b_K] \in \{0, 1\}^K$ where $b_k$ is a Bernoulli random variable and $K$ is the number of skill hyperplanes. We define the skill index $i = \sum_{k=1}^K 2^{k-1} b_k$ as a sum of Bernoulli random variables $b_k$. Note that this is but one approach to generate skills (and SPs). In principle this setup defines $2^K$ skills, but in practice, far fewer skills are typically used (see experiments). Furthermore, the complexity of the SP is governed by the VC-dimension. We can now define the probability of executing skill $i$ as a Bernoulli likelihood in Equation 1.

$$P(i|x, m) = P\left[i = \sum_{k=1}^K 2^{k-1} b_k\right] = \prod_k p_k(b_k = i_k|x, m) \ . \tag{1}$$

Here, $i_k \in \{0, 1\}$ is the value of the $k^{th}$ bit of $B$, $x$ is the current state and $m$ is a description of the MDP. The probability $p_k(b_k = 1|x, m)$ and $p_k(b_k = 0|x, m)$ are defined in Equation 2.

$$p_k(b_k = 1|x, m) = \frac{1}{1 + \exp(-\alpha \psi_{(x,m)}^T \beta_k)}, p_k(b_k = 0|x, m) = 1 - p_k(b_k = 1|x, m) \ . \tag{2}$$

We have made use of the logistic sigmoid function to ensure valid probabilities where $\psi_{x,m}^T \beta_k$ is a skill hyperplane and $\alpha > 0$ is a temperature parameter. The intuition here is that the $k^{th}$ bit of a skill, $b_k = 1$, if the skill hyperplane $\psi_{x,m}^T \beta_k > 0$ meaning that the skill's partition is in the positive half-space of the hyperplane. Similarly, $b_k = 0$ if $\psi_{x,m}^T \beta_k < 0$ corresponding to the negative half-space. Using skill 3 as an example with $K = 2$ hyperplanes in Figure 1a, we would define the Bernoulli likelihood of executing $\zeta_3$ as $p(i = 3|x, m) = p_1(b_1 = 1|x, m) \cdot p_2(b_2 = 1|x, m)$.

**Intra-Skill Policy:** Now that we can define the probability of executing a skill based on its SP, we define the intra-skill policy $\sigma_\theta$ for each skill. The Gibb's distribution is a commonly used function to define policies in RL (Sutton et al., 1999). Therefore we define the intra-skill policy for skill $i$, parameterized by $\theta_i \in \mathbb{R}^d$ as

$$\sigma_{\theta_i}(a|s) = \frac{\exp\left(\alpha \phi_{x,a}^T \theta_i\right)}{\sum_{b \in A} \exp\left(\alpha \phi_{x,b}^T \theta_i\right)} \ . \tag{3}$$

Here, $\alpha > 0$ is the temperature, $\phi_{x,a} \in \mathbb{R}^d$ is a feature vector that depends on the current state $x \in X$ and action $a \in A$. Now that we have a definition of both the probability of executing a skill and an intra-skill policy, we need to incorporate these distributions into the policy gradient setting using a *generalized* trajectory.

**Generalized Trajectory:** A generalized trajectory is necessary to derive policy gradient update rules with respect to the parameters $\Theta, \beta$ as will be shown in Section 4. A typical trajectory is usually defined as $\tau = (x_t, a_t, r_t, x_{t+1})_{t=0}^{T}$ where $T$ is the length of the trajectory. For a generalized trajectory, our algorithm emits a class $i_t$ at each timestep $t \geq 1$, which denotes the skill that was executed. The generalized trajectory is defined as $g = (x_t, a_t, i_t, r_t, x_{t+1})_{t=0}^{T}$. The probability of a generalized trajectory, as an extension to the PG trajectory in Section 2, is now, $P_{\Theta,\beta}(g) = P(x_0) \prod_{t=0}^{T} P(x_{t+1}|x_t, a_t) P_\beta(i_t|x_t, m) \sigma_{\theta_i}(a_t|x_t)$, where $P_\beta(i_t|x_t, m)$ is the probability of a skill being executed, given the state $x_t \in X$ and environment $m$ at time $t \geq 1$; $\sigma_{\theta_i}(a_t|x_t)$ is the probability of executing action $a_t \in A$ at time $t \geq 1$ given that we are executing skill $i$. The generalized trajectory is now a function of two parameter vectors $\theta$ and $\beta$.

# 4 Adaptive Skills, Adaptive Partitions (ASAP) Framework

The *Adaptive Skills, Adaptive Partitions (ASAP)* framework simultaneously learns a near-optimal set of skills and SPs (inter-skill policy), given an initially misspecified model. ASAP also automatically composes skills together and allows for a multi-task setting as it incorporates the environment $m$ into its hyperplane feature set. We have previously defined two important distributions $P_\beta(i_t|x_t, m)$ and $\sigma_{\theta_i}(a_t|x_t)$ respectively. These distributions are used to collectively define the ASAP policy which is presented below. Using the notion of a generalized trajectory, the ASAP policy can be learned in a policy gradient setting.

**ASAP Policy:** Assume that we are given a probability distribution $\mu$ over MDPs with a d-dimensional state-action space and a $z$-dimensional vector describing each MDP. We define $\beta$ as a $(d + z) \times K$ matrix where each column $\beta_i$ represents a skill hyperplane, and $\Theta$ is a $(d \times 2^K)$ matrix where each column $\theta_j$ parameterizes an intra-skill policy. Using the previously defined distributions, we now define the ASAP policy.

**Definition 3.** *(ASAP Policy). Given $K$ skill hyperplanes, a set of $2^K$ skills $\Sigma = \{\zeta_i | i = 1, \cdots 2^K\}$, a state space $x \in X$, a set of actions $a \in A$ and an MDP $m$ from a hypothesis space of MDPs, the ASAP policy is defined as,*

$$\pi_{\Theta,\beta}(a|x, m) = \sum_{i=1}^{2^K} P_\beta(i|x, m) \sigma_{\theta_i}(a|x) \ , \tag{4}$$

*where $P_\beta(i|x, m)$ and $\sigma_{\theta_i}(a|s)$ are the distributions as defined in Equations 1 and 3 respectively.*

This is a powerful description for a policy, which resembles a Bayesian approach, as the policy takes into account the uncertainty of the skills that are executing as well as the actions that each skill's intra-skill policy chooses. We now define the ASAP objective with respect to the ASAP policy.

**ASAP Objective:** We defined the policy with respect to a hypothesis space of $m$ MDPs. We now need to define an objective function which takes this hypothesis space into account. Since we assume that we are provided with a distribution $\mu : M \to [0, 1]$ over possible MDP models $m \in M$, with a $d$-dimensional state-action space, we can incorporate this into the ASAP objective function:

$$\rho(\pi_{\Theta,\beta}) = \int \mu(m) J^{(m)}(\pi_{\Theta,\beta}) dm \ , \tag{5}$$

where $\pi_{\Theta,\beta}$ is the ASAP policy and $J^{(m)}(\pi_{\Theta,\beta})$ is the expected return for MDP $m$ with respect to the ASAP policy. To simplify the notation, we group all of the parameters into a single parameter vector $\Omega = [vec(\Theta), vec(\beta)]$. We define the expected reward for *generalized trajectories* $g$ as $J(\pi_\Omega) = \int_g P_\Omega(g) R(g) dg$, where $R(g)$ is reward obtained for a particular trajectory $g$. This is a slight variation of the original policy gradient objective defined in Section 2. We then insert $J(\pi_\Omega)$ into Equation 5 and we get the ASAP objective function

$$\rho(\pi_\Omega) = \int \mu(m) J^{(m)}(\pi_\Omega) dm \ , \tag{6}$$

where $J^{(m)}(\pi_\Omega)$ is the expected return for policy $\pi_\Omega$ in MDP $m$. Next, we need to derive gradient update rules to learn the parameters of the optimal policy $\pi_\Omega^*$ that maximizes this objective.

**ASAP Gradients:** To learn both intra-skill policy parameters matrix $\Theta$ as well as the hyperplane parameters matrix $\beta$ (and therefore implicitly the SPs), we derive an update rule for the policy gradient framework with generalized trajectories. The full derivation is in the supplementary material. The first step involves calculating the gradient of the ASAP objective function yielding the ASAP gradient (Theorem 1).

**Theorem 1.** *(ASAP Gradient Theorem). Suppose that the ASAP objective function is $\rho(\pi_\Omega) = \int \mu(m) J^{(m)}(\pi_\Omega) dm$ where $\mu(m)$ is a distribution over MDPs $m$ and $J^{(m)}(\pi_\Omega)$ is the expected return for MDP $m$ whilst following policy $\pi_\Omega$, then the gradient of this objective is:*

$$\nabla_\Omega \rho(\pi_\Omega) = \mathbb{E}_{\mu(m)} \left[ \mathbb{E}_{P_\Omega^{(m)}(g)} \left[ \sum_{i=0}^{H^{(m)}} \nabla_\Omega Z_\Omega^{(m)}(x_t, i_t, a_t) R^{(m)} \right] \right] \ ,$$

*where $Z_\Omega^{(m)}(x_t, i_t, a_t) = \log P_\beta(i_t|x_t, m)\sigma_{\theta_i}(a_t|x_t)$, $H^{(m)}$ is the length of a trajectory for MDP $m$; $R^{(m)} = \sum_{i=0}^{H^{(m)}} \gamma^i r_i$ is the discounted cumulative reward for trajectory $H^{(m)}$ [1].*

If we are able to derive $\nabla_\Omega Z_\Omega^{(m)}(x_t, i_t, a_t)$, then we can estimate the gradient $\nabla_\Omega \rho(\pi_\Omega)$. We will refer to $Z_\Omega^{(m)} = Z_\Omega^{(m)}(x_t, i_t, a_t)$ where it is clear from context. It turns out that it is possible to derive this term as a result of the generalized trajectory. This yields the gradients $\nabla_\Theta Z_\Omega^{(m)}$ and $\nabla_\beta Z_\Omega^{(m)}$ in Theorems 2 and 3 respectively. The derivations can be found the supplementary material.

**Theorem 2.** *($\Theta$ Gradient Theorem). Suppose that $\Theta$ is a $(d \times 2^K)$ matrix where each column $\theta_j$ parameterizes an intra-skill policy. Then the gradient $\nabla_{\theta_{i_t}} Z_\Omega^{(m)}$ corresponding to the intra-skill parameters of the $i^{th}$ skill at time $t$ is:*

$$\nabla_{\theta_{i_t}} Z_\Omega^{(m)} = \alpha\phi_{x_t,a_t} - \frac{\alpha \left( \sum_{b \in A} \phi_{x_t,b_t} \exp(\alpha\phi_{x_t,b_t}^T \Theta_{i_t}) \right)}{\left( \sum_{b \in A} \exp(\alpha\phi_{x_t,b_t}^T \Theta_{i_t}) \right)} \ ,$$

*where $\alpha > 0$ is the temperature parameter and $\phi_{x_t,a_t} \in \mathbb{R}^{d \times 2^K}$ is a feature vector of the current state $x_t \in X_t$ and the current action $a_t \in A_t$.*

**Theorem 3.** *($\beta$ Gradient Theorem). Suppose that $\beta$ is a $(d + z) \times K$ matrix where each column $\beta_k$ represents a skill hyperplane. Then the gradient $\nabla_{\beta_k} Z_\Omega^{(m)}$ corresponding to parameters of the $k^{th}$ hyperplane is:*

$$\nabla_{\beta_{k,1}} Z_\Omega^{(m)} = \frac{\alpha\psi_{(x_t,m)} \exp(-\alpha\psi_{x_t,m}^T \beta_k)}{\left( 1 + \exp(-\alpha\psi_{x_t,m}^T \beta_k) \right)}, \nabla_{\beta_{k,0}} Z_\Omega^{(m)} = -\alpha\psi_{x_t,m} + \frac{\alpha\psi_{x_t,m} \exp(-\alpha\psi_{x_t,m}^T \beta_k)}{\left( 1 + \exp(-\alpha\psi_{x_t,m}^T \beta_k) \right)}$$
(7)

*where $\alpha > 0$ is the hyperplane temperature parameter, $\psi_{(x_t,m)}^T \beta_k$ is the $k^{th}$ skill hyperplane for MDP $m$, $\beta_{k,1}$ corresponds to locations in the binary vector equal to $1$ ($b_k = 1$) and $\beta_{k,0}$ corresponds to locations in the binary vector equal to $0$ ($b_k = 0$).*

Using these gradient updates, we can then order all of the gradients into a vector $\nabla_\Omega Z_\Omega^{(m)} = \langle \nabla_{\theta_1} Z_\Omega^{(m)} \ldots \nabla_{\theta_{2k}} Z_\Omega^{(m)}, \nabla_{\beta_1} Z_\Omega^{(m)} \ldots \nabla_{\beta_k} Z_\Omega^{(m)} \rangle$ and update both the intra-skill policy parameters and hyperplane parameters for the given task (learning a skill set and SPs). Note that the updates occur on a *single* time scale. This is formally stated in the ASAP Algorithm.

# 5  ASAP Algorithm

We present the ASAP algorithm (Algorithm 1) that dynamically and simultaneously learns skills, the inter-skill policy and automatically composes skills together by learning SPs. The skills ($\Theta$ matrix) and SPs ($\beta$ matrix) are initially arbitrary and therefore form a *misspecified model*. Line 2 combines

the skill and hyperplane parameters into a single parameter vector $\Omega$. Lines $3-7$ learns the skill and hyperplane parameters (and therefore implicitly the skill partitions). In line $4$ a generalized trajectory is generated using the current ASAP policy. The gradient $\nabla_\Omega \rho(\pi_\Omega)$ is then estimated in line $5$ from this trajectory and updates the parameters in line $6$. This is repeated until the skill and hyperplane parameters have converged, thus correcting the misspecified model. Theorem 4 provides a convergence guarantee of ASAP to a local optimum (see supplementary material for the proof).

---

**Algorithm 1** ASAP

---

**Require:** $\phi_{s,a} \in \mathcal{R}^d$ {state-action feature vector}, $\psi_{x,m} \in \mathcal{R}^{(d+z)}$ {skill hyperplane feature vector},
    $K$ {The number of hyperplanes}, $\Theta \in \mathbb{R}^{d \times 2^K}$ {An arbitrary skill matrix}, $\beta \in \mathbb{R}^{(d+z) \times K}$ {An arbitrary skill hyperplane matrix}, $\mu(m)$ {A distribution over MDP tasks}
1: $Z = (|d||2^K| + |(d+z)K|)$ {Define the number of parameters}
2: $\Omega = [vec(\Theta), vec(\beta)] \in \mathcal{R}^Z$
3: **repeat**
4: Perform a trial (which may consist of multiple MDP tasks) and obtain
    $x_{0:H}, i_{0:H}, a_{0:H}, r_{0:H}, m_{0:H}$ {states, skills, actions, rewards, task-specific information}
5: $\nabla_\Omega \rho(\pi_\Omega) = \left\langle \sum_m \left\langle \sum_{i=0}^{T^{(m)}} \nabla_\Omega Z^{(m)}(\Omega) R^{(m)} \right\rangle \right\rangle$ {$T$ is the task episode length}
6: $\Omega \to \Omega + \eta \nabla_\Omega \rho(\pi_\Omega)$
7: **until** parameters $\Omega$ have converged
8: **return** $\Omega$

---

**Theorem 4.** *Convergence of ASAP: Given an ASAP policy $\pi(\Omega)$, an ASAP objective over MDP models $\rho(\pi_\Omega)$ as well as the ASAP gradient update rules. If (1) the step-size $\eta_k$ satisfies $\lim_{k \to \infty} \eta_k = 0$ and $\sum_k \eta_k = \infty$; (2) The second derivative of the policy is bounded and we have bounded rewards; Then, the sequence $\{\rho(\pi_{\Omega,k})\}_{k=0}^{\infty}$ converges such that $\lim_{k \to \infty} \frac{\partial \rho(\pi_{\Omega,k})}{\partial \Omega} = 0$ almost surely.*

## 6 Experiments

The experiments have been performed on four different continuous domains: the *Two Rooms (2R)* domain (Figure 1$b$), the *Flipped 2R* domain (Figure 1$c$), the *Three rooms (3R)* domain (Figure 1$d$) and *RoboCup* domains (Figure 1$e$) that include a one-on-one scenario between a striker and a goalkeeper (R1), a two-on-one scenario of a striker against a goalkeeper and a defender (R2), and a striker against two defenders and a goalkeeper (R3) (see supplementary material). In each experiment, ASAP is provided with a *misspecified model*; that is, a set of skills and SPs (the inter-skill policy) that achieve degenerate, sub-optimal performance. ASAP corrects this misspecified model in each case to learn a set of near-optimal skills and SPs. For each experiment we implement ASAP using Actor-Critic Policy Gradient (AC-PG) as the learning algorithm [2].

**The Two-Room and Flipped Room Domains (2R):** In both domains, the agent (red ball) needs to reach the goal location (blue square) in the shortest amount of time. The agent receives constant negatives rewards and upon reaching the goal, receives a large positive reward. There is a wall dividing the environment which creates two rooms. The state space is a $4$-tuple consisting of the continuous $\langle x_{agent}, y_{agent} \rangle$ location of the agent and the $\langle x_{goal}, y_{goal} \rangle$ location of the center of the goal. The agent can move in each of the four cardinal directions. For each experiment involving the two room domains, a single hyperplane is learned (resulting in two SPs) with a linear feature vector representation $\psi_{x,m} = [1, x_{agent}, y_{agent}]$. In addition, a skill is learned in each of the two SPs. The intra-skill policies are represented as a probability distribution over actions.

**Automated Hyperplane and Skill Learning**: Using ASAP, the agent learned intuitive SPs and skills as seen in Figure 1$f$ and $g$. Each colored region corresponds to a SP. The white arrows have been superimposed onto the figures to indicate the skills learned for each SP. Since each intra-skill policy is a probability distribution over actions, each skill is unable to solve the entire task on its own. ASAP has taken this into account and has positioned the hyperplane accordingly such that the given skill representation can solve the task. Figure 2$a$ shows that ASAP improves upon the initial misspecified partitioning to attain near-optimal performance compared to executing ASAP on the fixed initial misspecified partitioning and on a fixed approximately optimal partitioning.

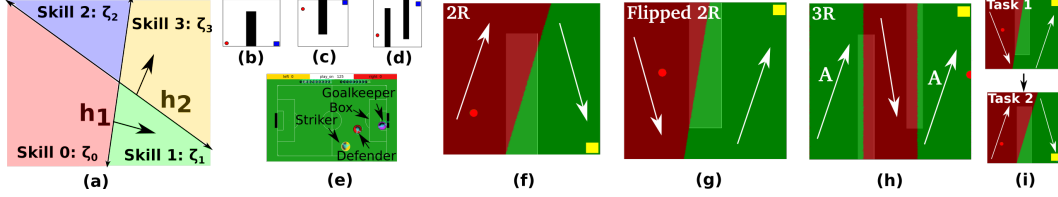

Figure 1: (a) The intersection of skill hyperplanes $\{h_1, h_2\}$ form four partitions, each of which defines a skill's execution region (the inter-skill policy). The (b) 2R, (c) Flipped 2R, (d) 3R and (e) RoboCup domains (with a varying number of defenders for R1,R2,R3). The learned skills and Skill Partitions (SPs) for the (f) 2R, (g) Flipped 2R, (h) 3R and (i) across multiple tasks.

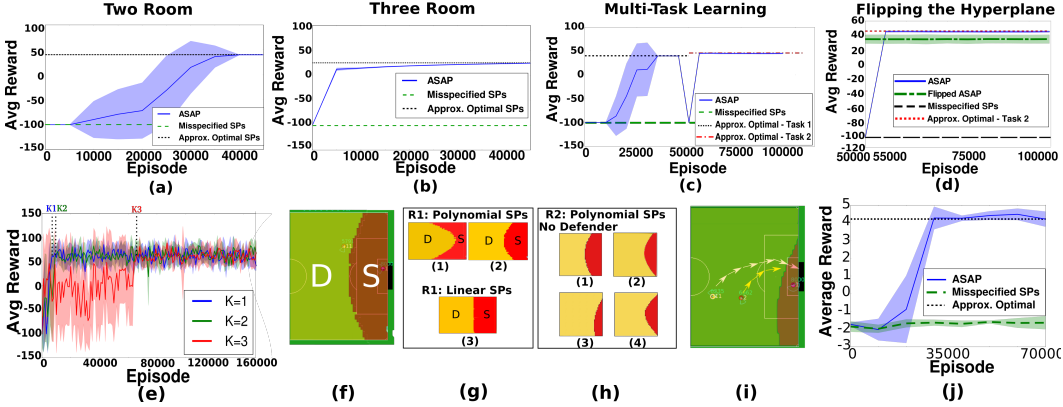

Figure 2: Average reward of the learned ASAP policy compared to (1) the approximately optimal SPs and skill set as well as (2) the initial misspecified model. This is for the (a) 2R, (b) 3R, (c) 2R learning across multiple tasks and the (d) 2R without learning by flipping the hyperplane. (e) The average reward of the learned ASAP policy for a varying number of $K$ hyperplanes. (f) The learned SPs and skill set for the R1 domain. (g) The learned SPs using a polynomial hyperplane (1),(2) and linear hyperplane (3) representation. (h) The learned SPs using a polynomial hyperplane representation without the defender's location as a feature (1) and with the defender's $x$ location (2), $y$ location (3), and $\langle x, y \rangle$ location as a feature (4). (i) The dribbling behavior of the striker when taking the defender's $y$ location into account. (j) The average reward for the R1 domain.

**Multiple Hyperplanes:** We analyzed the ASAP framework when learning multiple hyperplanes in the two room domain. As seen in Figure 2e, increasing the number of hyperplanes $K$, does not have an impact on the final solution in terms of average reward. However, it does increase the computational complexity of the algorithm since $2^K$ skills need to be learned. The approximate points of convergence are marked in the figure as $K1, K2$ and $K3$, respectively. In addition, two skills dominate in each case producing similar partitions to those seen in Figure 1a (see supplementary material) indicating that ASAP learns that not all skills are necessary to solve the task.

**Multitask Learning**: We first applied ASAP to the 2R domain (Task 1) and attained a near optimal average reward (Figure 2c). It took approximately 35000 episodes to get near-optimal performance and resulted in the SPs and skill set shown in Figure 1i (top). Using the learned SPs and skills, ASAP was then able to adapt and learn a new set of SPs and skills to solve a different task (Flipped 2R - Task 2) in only 5000 episodes (Figure 2c) indicating that the parameters learned from the old task provided a good initialization for the new task. The knowledge transfer is seen in Figure 1i (bottom) as the SPs do not significantly change between tasks, yet the skills are completely relearned.

We also wanted to see whether we could *flip* the SPs; that is, switch the sign of the hyperplane parameters learned in the 2R domain and see whether ASAP can solve the Flipped 2R domain (Task 2) **without** any additional learning. Due to the symmetry of the domains, ASAP was indeed able to solve the new domain and attained near-optimal performance (Figure 2d). This is an exciting result as many problems, especially navigation tasks, possess symmetrical characteristics. This insight could dramatically reduce the sample complexity of these problems.

**The Three-Room Domain (3R):** The 3R domain (Figure 1d), is similar to the 2R domain regarding the goal, state-space, available actions and rewards. However, in this case, there are two walls, dividing the state space into three rooms. The hyperplane feature vector $\psi_{x,m}$ consists of a single

fourier feature. The intra-skill policy is a probability distribution over actions. The resulting learned hyperplane partitioning and skill set are shown in Figure $1h$. Using this partitioning ASAP achieved near optimal performance (Figure $2b$). This experiment shows an insightful and unexpected result. **Reusable Skills**: Using this hyperplane representation, ASAP was able to not only learn the intra-skill policies and SPs, but also that skill 'A' needed to be *reused* in two different parts of the state space (Figure $1h$). ASAP therefore shows the potential to automatically create reusable skills.

**RoboCup Domain:** The RoboCup 2D soccer simulation domain (Akiyama & Nakashima, 2014) is a 2D soccer field (Figure $1e$) with two opposing teams. We utilized three RoboCup sub-domains [3] R1, R2 and R3 as mentioned previously. In these sub-domains, a striker (the agent) needs to learn to dribble the ball and try and score goals past the goalkeeper. **State space:** R1 domain - the continuous locations of the striker $\langle x_{striker}, y_{striker} \rangle$ , the ball $\langle x_{ball}, y_{ball} \rangle$, the goalkeeper $\langle x_{goalkeeper}, y_{goalkeeper} \rangle$ and the constant goal location $\langle x_{goal}, y_{goal} \rangle$. R2 domain - we have the addition of the defender's location $\langle x_{defender}, y_{defender} \rangle$ to the state space. R3 domain - we add the locations of two defenders. **Features:** For the R1 domain, we tested both a linear and degree two polynomial feature representation for the hyperplanes. For the R2 and R3 domains, we also utilized a degree two polynomial hyperplane feature representation. **Actions:** The striker has three actions which are (1) move to the ball (**M**), (2) move to the ball and dribble towards the goal (**D**) (3) move to the ball and shoot towards the goal (**S**). **Rewards:** The reward setup is consistent with logical football strategies (Hausknecht & Stone, 2015; Bai et al., 2012). Small negative (positive) rewards for shooting from outside (inside) the box and dribbling when inside (outside) the box. Large negative rewards for losing possession and kicking the ball out of bounds. Large positive reward for scoring.

**Different SP Optimas**: Since ASAP attains a locally optimal solution, it may sometimes learn different SPs. For the polynomial hyperplane feature representation, ASAP attained two different solutions as shown in Figure $2g(1)$ as well as Figure $2g(2)$, respectively. Both achieve near optimal performance compared to the approximately optimal scoring controller (see supplementary material). For the linear feature representation, the SPs and skill set in Figure $2g(3)$ is obtained and achieved near-optimal performance (Figure $2j$), outperforming the polynomial representation.

**SP Sensitivity**: In the R2 domain, an additional player (the defender) is added to the game. It is expected that the presence of the defender will affect the shape of the learned SPs. ASAP again learns intuitive SPs. However, the shape of the learned SPs change based on the pre-defined hyperplane feature vector $\psi_{m,x}$. Figure $2h(1)$ shows the learned SPs when the location of the defender is *not* used as a hyperplane feature. When the $x$ location of the defender is utilized, the 'flatter' SPs are learned in Figure $2h(2)$. Using the $y$ location of the defender as a hyperplane feature causes the hyperplane offset shown in Figure $2h(3)$. This is due to the striker learning to dribble around the defender in order to score a goal as seen in Figure $2i$. Finally, taking the $\langle x, y \rangle$ location of the defender into account results in the 'squashed' SPs shown in Figure $2h(4)$ clearly showing the sensitivity and adaptability of ASAP to dynamic factors in the environment.

# 7 Discussion

We have presented the Adaptive Skills, Adaptive Partitions (ASAP) framework that is able to automatically compose skills together and learns a near-optimal skill set and skill partitions (the inter-skill policy) simultaneously to correct an initially misspecified model. We derived the gradient update rules for both skill and skill hyperplane parameters and incorporated them into a policy gradient framework. This is possible due to our definition of a generalized trajectory. In addition, ASAP has shown the potential to learn across multiple tasks as well as automatically reuse skills. These are the necessary requirements for a truly general skill learning framework and can be applied to lifelong learning problems (Ammar et al., 2015; Thrun & Mitchell, 1995). An exciting extension of this work is to incorporate it into a Deep Reinforcement Learning framework, where both the skills and ASAP policy can be represented as deep networks.

**Acknowledgements**

The research leading to these results has received funding from the European Research Council under the European Union's Seventh Framework Program (FP/2007-2013) / ERC Grant Agreement n. 306638.

## Footnotes

[1]These expectations can easily be sampled (see supplementary material).

[2] AC-PG works well in practice and can be trivially incorporated into ASAP with convergence guarantees

[3]https://github.com/mhauskn/HFO.git

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
