[Supplementary Material · asap_supp.pdf]

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

$$
\begin{aligned}
&\nabla_\Omega \rho(\pi_\Omega) \\
=\ & \int \mu(m) \nabla_\Omega J^{(m)}(\pi_\Omega) dm \\
=\ & \int \mu(m) \nabla_\Omega \left(\int_{g^{(m)}} P_\Omega^{(m)}(g) R^{(m)}(g) dg\right) dm \\
=\ & \int \mu(m) \left(\int_{g^{(m)}} \nabla_\Omega P_\Omega^{(m)}(g) R^{(m)}(g) dg\right) dm \\
=\ & \int \mu(m) \left(\int_{g^{(m)}} P_\Omega^{(m)}(g) \nabla_\Omega \log P_\Omega^{(m)}(g) R^{(m)}(g) dg\right) dm \\
=\ & \int \mu(m) \left(\int_{g^{(m)}} P_\Omega^{(m)}(g) \nabla_\Omega \log P_\Omega^{(m)}(g) R^{(m)}(g) dg\right) dm \\
=\ & \int \mu(m) \mathbb{E}_{P_\Omega^{(m)}(g)}\left[\nabla_\Omega \log P_\Omega^{(m)}(g) R^{(m)}(g)\right] dm \\
=\ & \int \mu(m) \mathbb{E}_{P_\Omega^{(m)}(g)}\left[\sum_{t=0}^{H^{(m)}} \nabla_\Omega \log P_\beta(i_t|x_t, m)\sigma_{\theta_{i_t}}(a_t|x_t) \sum_{j=0}^{H^{(m)}} \gamma^j r_j\right] dm \\
=\ & \mathbb{E}_{\mu(m)}\left[\mathbb{E}_{P_\Omega^{(m)}(g)}\left[\sum_{i=0}^{H^{(m)}} \nabla_\Omega Z_\Omega^{(m)}(x_t, i_t, a_t) \sum_{j=0}^{H^{(m)}} \gamma^j r_j\right]\right]
\end{aligned}
$$

$\square$

The double expectation can be sampled as follows:

$$\nabla_\Omega \rho(\pi_\Omega)$$

$$= \mathbb{E}_{\mu(m)} \left[ \left\langle \sum_{t=0}^{H^{(m)}} \nabla_\Omega Z_\Omega^{(m)}(x_t, i_t, a_t) \sum_{j=0}^{H^{(m)}} \gamma^j r_j \right\rangle \right]$$

$$= \mathbb{E}_{\mu(m)} \left[ \left\langle \sum_{t=0}^{H^{(m)}} \nabla_\Omega Z_\Omega^{(m)}(x_t, i_t, a_t) \sum_{j=0}^{H^{(m)}} \gamma^j r_j \right\rangle \right]$$

$$= \left\langle \sum_m \left\langle \sum_{i=0}^{H^{(m)}} \nabla_\Omega Z_\Omega^{(m)}(x_t, i_t, a_t) \sum_{j=0}^{H^{(m)}} \gamma^j r_j \right\rangle \right\rangle ,$$

where $\left\langle \cdot \right\rangle$ represent an average over trajectories. If we are able to derive $\nabla_\Omega Z_\Omega^{(m)}(x_t, i_t, a_t)$, then we can estimate the gradient $\nabla_\Omega \rho(\pi_\Omega)$. We will refer to $Z_\Omega^{(m)} = Z_\Omega^{(m)}(x_t, i_t, a_t)$ where it is clear from context. It turns out that it is possible to derive this term as a result of the generalized trajectory. This yields the gradients $\nabla_\Theta Z_\Omega^{(m)}$ and $\nabla_\beta Z_\Omega^{(m)}$ in Theorems 2 and 3 respectively. The derivations can be found the supplementary material.

**Theorem 2.** *($\Theta$ Gradient Theorem). Suppose that $\Theta$ is a $(d \times 2^K)$ matrix where each column $\theta_j$ parameterizes an intra-skill policy. Then the gradient $\nabla_{\theta_{i_t}} Z_\Omega^{(m)}$ corresponding to the intra-skill parameters of the $i^{th}$ skill at time t is:*

$$\nabla_{\theta_{i_t}} Z_\Omega^{(m)} = \alpha \phi_{x_t, a_t} - \frac{\alpha \left( \sum_{b \in A} \phi_{x_t, b_t} \exp(\alpha \phi_{x_t, b_t}^T \Theta_{i_t}) \right)}{\left( \sum_{b \in A} \exp(\alpha \phi_{x_t, b_t}^T \Theta_{i_t}) \right)} ,$$

*where $\alpha > 0$ is the temperature parameter and $\phi_{x_t, a_t} \in \mathbb{R}^{d \times 2^K}$ is a feature vector of the current state $x_t \in X_t$ and the current action $a_t \in A_t$.*

**Theorem 3.** *($\beta$ Gradient Theorem). Suppose that $\beta$ is a $(d + z) \times K$ matrix where each column $\beta_k$ represents a skill hyperplane. Then the gradient $\nabla_{\beta_k} Z_\Omega^{(m)}$ corresponding to parameters of the $k^{th}$ hyperplane is:*

$$\nabla_{\beta_{k,1}} Z_\Omega^{(m)} = \frac{\alpha \psi_{(x_t, m)} \exp(-\alpha \psi_{x_t, m}^T \beta_k)}{\left( 1 + \exp(-\alpha \psi_{x_t, m}^T \beta_k) \right)}, \nabla_{\beta_{k,0}} Z_\Omega^{(m)} = -\alpha \psi_{x_t, m} + \frac{\alpha \psi_{x_t, m} \exp(-\alpha \psi_{x_t, m}^T \beta_k)}{\left( 1 + \exp(-\alpha \psi_{x_t, m}^T \beta_k) \right)}$$

$$(7)$$

*where $\alpha > 0$ is the hyperplane temperature parameter, $\psi_{(x_t, m)}^T \beta_k$ is the $k^{th}$ skill hyperplane for MDP m, $\beta_{k,1}$ corresponds to locations in the binary vector equal to 1 ($b_k = 1$) and $\beta_{k,0}$ corresponds to locations in the binary vector equal to 0 ($b_k = 0$).*

*Proof.* First let us define a function $\chi(\cdot)$. Given a value $l \in L$, the function $\chi(l) \rightarrow j = Loc(Bi(l)) \in \mathbb{R}^Q$. Here $Bi(l)$ is a function that maps $l$ to a binary vector of value equal to $l$ and $Loc$ indicates the $Q$ indices in the binary vector where the corresponding elements are equal to 1.

Since $P_\beta(i_t | x_t, m)$ is defined as a Bernoulli likelihood distribution and $\sigma_{\theta_i}(a_t | x_t)$ is defined as a Gibb's distribution, the product $P_\beta(i_t | x_t, m) \cdot \sigma_{\theta_i}(a_t | x_t)$ can be defined in Equation 8.

$$P_\beta(i_t | x_t, m) \sigma_{\theta_i}(a_t | x_t)$$

$$= \frac{\exp\left(\alpha \phi_{x_t, a_t}^T \Theta_{i_t}\right)}{\sum_{b \in A} \exp\left(\alpha \phi_{x_t, b_t}^T \Theta_{i_t}\right)} \prod_k P_{k,\beta}(b_k = i_k | x, m)$$

$$= \frac{\exp\left(\alpha \phi_{x_t, a_t}^T \Theta_{i_t}\right)}{\sum_{b \in A} \exp\left(\alpha \phi_{x_t, b_t}^T \Theta_{i_t}\right)} \left( \prod_{l=\chi(i)} \frac{1}{1 + \exp(-\alpha \psi_{(x_t, m)}^T \beta_l)} \right) \left( \prod_{j \neq \chi(i)} \frac{\exp(-\alpha \psi_{(x_t, m)}^T \beta_j)}{1 + \exp(-\alpha \psi_{(x_t, m)}^T \beta_j)} \right)$$

Now, taking the log of both sides, yields:

$$
\begin{aligned}
& Z_\Omega^{(m)}(x_t, i_t, a_t) \\
=\ & \log P_\beta(i_t|x_t, m) + \log \sigma_{\theta_{i_t}}(a_t|s_t) \\
=\ & \log\left(\frac{\exp\left(\alpha\phi_{x_t,a_t}^T\Theta_{i_t}\right)}{\sum_{b\in A}\exp\left(\alpha\phi_{x_t,b_t}^T\Theta_{i_t}\right)}\right) + \log\left(\prod_k p_k(b_k = i_k|x, m)\right) \\
=\ & \log\left(\frac{\exp\left(\alpha\phi_{x_t,a_t}^T\Theta_{i_t}\right)}{\sum_{b\in A}\exp\left(\alpha\phi_{x_t,b_t}^T\Theta_{i_t}\right)}\right) + \log\left[\left(\prod_{j=\chi(i)}\frac{1}{1+\exp(-\alpha\psi_{(x_t,m)}^T\beta_j)}\right)\right. \\
& \left.\left(\prod_{k\neq\chi(i)}\frac{\exp(-\alpha\psi_{(x_t,m)}^T\beta_k)}{1+\exp(-\alpha\psi_{(x_t,m)}^T\beta_k)}\right)\right] \\
=\ & \log\left(\frac{\exp\left(\alpha\phi_{x_t,a_t}^T\Theta_{i_t}\right)}{\sum_{b\in A}\exp\left(\alpha\phi_{x_t,b_t}^T\Theta_{i_t}\right)}\right) + \log\left(\prod_{j=\chi(i)}\frac{1}{1+\exp(-\alpha\psi_{(x_t,m)}^T\beta_j)}\right) \\
& +\ \log\left(\prod_{k\neq\chi(i)}\frac{\exp(-\alpha\psi_{(x_t,m)}^T\beta_k)}{1+\exp(-\alpha\psi_{(x_t,m)}^T\beta_k)}\right) \\
=\ & \alpha\phi_{x_t,a_t}^T\Theta_{i_t} - \log\left(\sum_{b\in A}\exp(\alpha\phi_{x_t,b_t}^T\Theta_{i_t})\right) + \sum_{j=\chi(i)}\log\left(\frac{1}{1+\exp(-\alpha\psi_{(x_t,m)}^T\beta_j)}\right) \\
& +\ \sum_{k\neq\chi(i)}\log\left(\frac{\exp(-\alpha\psi_{(x_t,m)}^T\beta_k)}{1+\exp(-\alpha\psi_{(x_t,m)}^T\beta_k)}\right) \\
=\ & \alpha\phi_{x_t,a_t}^T\Theta_{i_t} - \log\left(\sum_{b\in A}\exp(\alpha\phi_{x_t,b_t}^T\Theta_{i_t})\right) - \sum_{j=\chi(i)}\log\left(1+\exp(-\alpha\psi_{(x_t,m)}^T\beta_j)\right) \\
& +\ \sum_{k\neq\chi(i)}-\alpha\psi_{(x_t,m)}^T\beta_k - \log\left(1+\exp(-\alpha\psi_{(x_t,m)}^T\beta_k)\right)
\end{aligned}
$$

Using this result, we derive the gradient $\nabla_\Omega Z_\Omega^{(m)}(x_t, i_t, a_t)$ with respect to each of the parameters $\Omega$. We first start by taking the gradient with respect to $\theta_{i_t,1}$ which corresponds to the first element of skill $i$'s intra-skill policy parameter vector at time $t$. We have,

$$
\begin{aligned}
& \frac{\partial}{\partial\theta_{i_t,1}}Z_\Omega^{(m)}(x_t, i_t, a_t) \\
=\ & \frac{\partial}{\partial\theta_{i_t,1}}\left[\alpha\phi_{x_t,a_t}^T\Theta_{i_t} - \log\left(\sum_{b\in A}\exp(\alpha\phi_{x_t,b_t}^T\Theta_{i_t})\right) - \sum_{j=\chi(i)}\log\left(1+\exp(-\alpha\psi_{(x_t,m)}^T\beta_j)\right)\right. \\
& +\ \left.\sum_{k\neq\chi(i)}-\alpha\psi_{(x_t,m)}^T\beta_k - \log\left(1+\exp(-\alpha\psi_{(x_t,m)}^T\beta_k)\right)\right] \\
=\ & \frac{\partial}{\partial\theta_{i_t,1}}\left[\alpha\phi_{x_t,a_t}^T\Theta_{i_t} - \log\left(\sum_{b\in A}\exp(\alpha\phi_{x_t,b_t}^T\Theta_{i_t})\right)\right] \\
=\ & \alpha\phi_{x_t,a_t,1} - \frac{\alpha\left(\sum_{b\in A}\phi_{x_t,b_t,1}\exp(\alpha\phi_{x_t,b_t}^T\Theta_{i_t})\right)}{\left(\sum_{b\in A}\exp(\alpha\phi_{x_t,b_t}^T\Theta_{i_t})\right)}
\end{aligned}
$$

So, the derivative with respect to $\theta_{i_t}$ is:

$$\nabla_{\theta_{i_t}} Z_{\Omega}^{(m)}(x_t, i_t, a_t)$$

$$= \alpha\phi_{x_t,a_t} - \frac{\alpha\left(\sum_{b\in A} \phi_{x_t,b_t} \exp(\alpha\phi_{x_t,b_t}^T \Theta_{i_t})\right)}{\left(\sum_{b\in A} \exp(\alpha\phi_{x_t,b_t}^T \Theta_{i_t})\right)} \tag{8}$$

The next step is to derive the gradient with respect to the parameters $\beta$ corresponding to the hyperplane parameters. We start with $\beta_{j1,1}$, the first element of the parameter vector for hyperplane $j1$. The location in the binary vector corresponding to $j1$ is $b_{j1} = 1$ meaning that hyperplane $j1$ generates a value of 1 when skill $i_t$ is being executed.

$$\frac{\partial}{\partial \beta_{j1,1}} Z_{\Omega}^{(m)}(x_t, i_t, a_t)$$

$$= \frac{\partial}{\partial \beta_{j1,1}} \left[ \alpha\phi_{x_t,a_t}^T \Theta_{i_t} - \log\left(\sum_{b\in A} \exp(\alpha\phi_{x_t,b_t}^T \Theta_{i_t})\right) - \sum_{j=\chi(i)} \log\left(1 + \exp(-\alpha\psi_{(x_t,m)}^T \beta_j)\right) \right.$$

$$+ \left. \sum_{k\neq\chi(i)} -\alpha\psi_{(x_t,m)}^T \beta_k - \log\left(1 + \exp(-\alpha\psi_{(x_t,m)}^T \beta_k)\right) \right]$$

$$= \frac{\partial}{\partial \beta_{j1,1}} \left[ -\sum_{j=\chi(i)} \log\left(1 + \exp(-\alpha\psi_{(x_t,m)}^T \beta_j)\right) + \sum_{k\neq\chi(i)} -\alpha\psi_{(x_t,m)}^T \beta_k - \log\left(1 + \exp(-\alpha\psi_{(x_t,m)}^T \beta_k)\right) \right]$$

$$= \frac{\alpha\psi_{(x_t,m,1)} \exp(-\alpha\psi_{(x_t,m)}^T \beta_{j1})}{\left(1 + \exp(-\alpha\psi_{(x_t,m)}^T \beta_{j1})\right)}$$

Therefore, the gradient of hyperplane $j1$ is defined as:

$$\nabla_{\beta_{j1,1}} Z_{\Omega}^{(m)}(x_t, i_t, a_t) = \nabla_{\beta_{(j1,b_{j1}=1)}} Z_{\Omega}^{(m)}(x_t, i_t, a_t)$$

$$= \frac{\alpha\psi_{(x_t,m)} \exp(-\alpha\psi_{(x_t,m)}^T \beta_{j1})}{\left(1 + \exp(-\alpha\psi_{(x_t,m)}^T \beta_{j1})\right)} \tag{9}$$

We then also need to derive with respect to the parameters $\beta_{k1,1}$, the first element of hyperplane $k1$. The location in the binary vector corresponding to $k1$ is $b_{k1} = 0$. Therefore, hyperplane $k1$ generates a value of 0 when skill $i_t$ is being executed.

$$\frac{\partial}{\partial \beta_{k1,1}} Z_{\Omega}^{(m)}(x_t, i_t, a_t)$$

$$= \frac{\partial}{\partial \beta_{k1,1}} \left[ \alpha\phi_{x_t,a_t}^T \Theta_{i_t} - \log\left(\sum_{b\in A} \exp(\alpha\phi_{x_t,b_t}^T \Theta_{i_t})\right) - \sum_{j=\chi(i)} \log\left(1 + \exp(-\alpha\psi_{(x_t,m)}^T \beta_j)\right) \right.$$

$$+ \left. \sum_{k\neq\chi(i)} -\alpha\psi_{(x_t,m)}^T \beta_k - \log\left(1 + \exp(-\alpha\psi_{(x_t,m)}^T \beta_k)\right) \right]$$

$$= \frac{\partial}{\partial \beta_{k1,1}} \left[ \sum_{k\neq\chi(i)} -\alpha\psi_{(x_t,m)}^T \beta_k - \log\left(1 + \exp(-\alpha\psi_{(x_t,m)}^T \beta_k)\right) \right]$$

$$= -\alpha\psi_{(x_t,m,1)} + \frac{\alpha\psi_{(x_t,m,1)} \exp(-\alpha\psi_{(x_t,m)}^T \beta_{k1})}{\left(1 + \exp(-\alpha\psi_{(x_t,m)}^T \beta_{k1})\right)}$$

The gradient is therefore,

$$\nabla \beta_{k1,0} Z_{\Omega}^{(m)}(x_t, i_t, a_t) = \nabla_{\beta_{(k1, b_{k1}=0)}} Z_{\Omega}^{(m)}(x_t, i_t, a_t)$$

$$= -\alpha \psi_{(x_t,m)} + \frac{\alpha \psi_{(x_t,m)} \exp(-\alpha \psi_{(x_t,m)}^T \beta_{k1})}{\left(1 + \exp(-\alpha \psi_{(x_t,m)}^T \beta_{k1})\right)} \tag{10}$$

The overall gradient is therefore

$$\nabla \theta_{i_t} Z_{\Omega}^{(m)}(x_t, i_t, a_t)$$

$$= \alpha \phi_{x_t,a_t} - \frac{\alpha \left( \sum_{b \in A} \phi_{x_t,b_t} \exp(\alpha \phi_{x_t,b_t}^T \Theta_{i_t}) \right)}{\left( \sum_{b \in A} \exp(\alpha \phi_{x_t,b_t}^T \Theta_{i_t}) \right)}$$

$$\nabla \beta_{k,1} = \nabla \beta_{(k,b_k=1)} Z_{\Omega}^{(m)}(x_t, i_t, a_t)$$

$$= \frac{\alpha \psi_{(x_t,m)} \exp(-\alpha \psi_{(x_t,m)}^T \beta_k)}{\left(1 + \exp(-\alpha \psi_{(x_t,m)}^T \beta_k)\right)}$$

$$\nabla \beta_{k,0} Z_{\Omega}^{(m)}(x_t, i_t, a_t) = \nabla \beta_{(k,b_K=0)} Z_{\Omega}^{(m)}(x_t, i_t, a_t)$$

$$= -\alpha \psi_{(x_t,m)} + \frac{\alpha \psi_{(x_t,m)} \exp(-\alpha \psi_{(x_t,m)}^T \beta_k)}{\left(1 + \exp(-\alpha \psi_{(x_t,m)}^T \beta_k)\right)}$$

Further simplifications yields:

$$\nabla \theta_{i_t} Z_{\Omega}^{(m)}(x_t, i_t, a_t)$$

$$= \alpha \phi_{x_t,a_t} - \alpha \phi_{x_t,a_t} \pi(x_t, a_t) \tag{11}$$

$$\nabla \beta_{(k,b_k=1)} Z_{\Omega}^{(m)}(x_t, i_t, a_t)$$

$$= \frac{\alpha \psi_{(x_t,m)} \exp(-\alpha \psi_{(x_t,m)}^T \beta_k)}{\left(1 + \exp(-\alpha \psi_{(x_t,m)}^T \beta_k)\right)} \tag{12}$$

$$\nabla \beta_{k,0} Z_{\Omega}^{(m)}(x_t, i_t, a_t) = \nabla \beta_{(k,b_k=0)} Z_{\Omega}^{(m)}(x_t, i_t, a_t)$$

$$= -\alpha \psi_{(x_t,m)} + \frac{\alpha \psi_{(x_t,m)} \exp(-\alpha \psi_{(x_t,m)}^T \beta_k)}{\left(1 + \exp(-\alpha \psi_{(x_t,m)}^T \beta_k)\right)} \tag{13}$$

$\square$

Using these gradient updates, we can then order all of the gradients into a vector $\nabla_{\Omega} Z_{\Omega}^{(m)} = \langle \nabla_{\theta_1} Z_{\Omega}^{(m)} \dots \nabla_{\theta_{2k}} Z_{\Omega}^{(m)}, \nabla_{\beta_1} Z_{\Omega}^{(m)} \dots \nabla_{\beta_k} Z_{\Omega}^{(m)} \rangle$ and update both the intra-skill policy parameters and hyperplane parameters for the given task (learning a skill set and SPs). Note that the updates occur on a *single* time scale. This is formally stated in the ASAP Algorithm.

# 5 ASAP Algorithm

We present the ASAP algorithm (Algorithm 1) that dynamically and simultaneously learns skills, the inter-skill policy and automatically composes skills together by learning SPs. The skills ($\Theta$ matrix) and SPs ($\beta$ matrix) are initially arbitrary and therefore form a *misspecified model*. Line 2 combines the skill and hyperplane parameters into a single parameter vector $\Omega$. Lines $3 - 7$ learns the skill and hyperplane parameters (and therefore implicitly the skill partitions). In line 4 a generalized trajectory is generated using the current ASAP policy. The gradient $\nabla_\Omega \rho(\pi_\Omega)$ is then estimated in line 5 from this trajectory and updates the parameters in line 6. This is repeated until the skill and hyperplane parameters have converged, thus correcting the misspecified model. Theorem 4 provides a convergence guarantee of ASAP to a local optimum (see supplementary material for the proof).

---

**Algorithm 1** ASAP

---

**Require:** $\phi_{s,a} \in \mathcal{R}^d$ {state-action feature vector}, $\psi_{x,m} \in \mathcal{R}^{(d+z)}$ {skill hyperplane feature vector},
    $K$ {The number of hyperplanes}, $\Theta \in \mathbb{R}^{d \times 2^K}$ {An arbitrary skill matrix}, $\beta \in \mathbb{R}^{(d+z) \times K}$ {An
    arbitrary skill hyperplane matrix}, $\mu(m)$ {A distribution over MDP tasks}
1: $Z = (|d||2^K| + |(d+z)K|)$ {Define the number of parameters}
2: $\Omega = [vec(\Theta), vec(\beta)] \in \mathcal{R}^Z$
3: **repeat**
4: Perform a trial (which may consist of multiple MDP tasks) and obtain
    $x_{0:H}, i_{0:H}, a_{0:H}, r_{0:H}, m_{0:H}$ {states, skills, actions, rewards, task-specific information}
5: $\nabla_\Omega \rho(\pi_\Omega) = \left\langle \sum_m \left\langle \sum_{i=0}^{T^{(m)}} \nabla_\Omega Z^{(m)}(\Omega) R^{(m)} \right\rangle \right\rangle$ {$T$ is the task episode length}
6: $\Omega \rightarrow \Omega + \eta \nabla_\Omega \rho(\pi_\Omega)$
7: **until** parameters $\Omega$ have converged
8: **return** $\Omega$

---

**Theorem 4.** *Convergence of ASAP: Given an ASAP policy $\pi(\Omega)$, an ASAP objective over MDP models $\rho(\pi_\Omega)$ as well as the ASAP gradient update rules. If (1) the step-size $\eta_k$ satisfies $\lim_{k \to \infty} \eta_k = 0$ and $\sum_k \eta_k = \infty$; (2) The second derivative of the policy is bounded and we have bounded rewards; Then, the sequence $\{\rho(\pi_{\Omega,k})\}_{k=0}^\infty$ converges such that $\lim_{k \to \infty} \frac{\partial \rho(\pi_{\Omega,k})}{\partial \Omega} = 0$ almost surely.*

*Proof.* We will now show convergence of the value function to a local minimum. As shown in Sutton et al. (2000), the gradient of the objective function can be represented as shown in Equation 14.

$$\frac{\partial \rho}{\partial \Omega} = \sum_x d^\pi(x) \sum_a \frac{\partial \pi_\Omega(x,a)}{\partial \Omega} Q^\pi(x,a) \ . \tag{14}$$

This is shown for both the long-term reward formulation for a given starting state $x_0$ and for the average reward formulation respectively. We will adapt our proof to the starting state formulation (noting that the average reward formulation can also be derived in a similar manner). Our proof needs to take the MDP models into account as shown by our objective in Equation 15. The objective function is defined for a discrete set of models, but can also be adapted to a continuous set.

$$\rho(\pi_\Omega) = \sum_{m \in M} \mu(m) J^{(m)}(\pi_\Omega) \ , \tag{15}$$

where $\Omega$ is the set of parameters for both the skills and hyperplanes respectively. In order to adapt our setting to that of Sutton et al. (2000), we start by deriving our objective with respect to the parameters as shown in Equation 17.

$$\rho(\pi_\Omega) \quad = \quad \sum_{m \in M} \mu(m) J^{(m)}(\pi_\Omega) \tag{16}$$

$$\frac{\partial \rho(\pi_\Omega)}{\partial \Omega} \quad = \quad \frac{\partial}{\partial \Omega} \sum_{m \in M} \mu(m) J^{(m)}(\pi_\Omega) \tag{17}$$

$$= \quad \sum_{m \in M} \mu(m) \frac{\partial}{\partial \Omega} J^{(m)}(\pi_\Omega)$$

Now, we need to analyze the derivative with respect to the parameters $\Omega$ for the value function $J^{(m)}(\pi_\Omega)$ for a model $m$. As in Sutton et al. (2000), the discounted weighting of states for a particular model $m$ is given by $d_{(m)}^{\pi}(x) = \sum_{t=0}^{\infty} \gamma^t P^{(m)}(x_t \to x | x_0, \pi)$. Using this observation, we can then derive the gradient $\frac{\partial}{\partial \Omega} J^{(m)}(\pi_\Omega)$ as shown in the equation below.

$$J^{(m)}(\pi_\Omega)|_{x_0 = x} = V_{(m)}^{\pi_\Omega}(x) = \sum_a \pi_\Omega^{(m)}(x, a) Q_{(m)}^{\pi_\Omega}(x, a)$$

$$\frac{\partial J^{(m)}(\pi_\Omega)|_{x_0 = x}}{\partial \Omega} = \sum_a \frac{\partial}{\partial \Omega} \pi_\Omega^{(m)}(x, a) Q_{(m)}^{\pi_\Omega}(x, a)$$

$$\Updownarrow$$

$$= \sum_y \sum_{k=0}^{\infty} \gamma^k P^{(m)}(s \to y, k, \pi) \sum_a \frac{\partial \pi_\Omega^{(m)}(x, a)}{\partial \Omega} Q_{(m)}^{\pi_\Omega}(x, a)$$

$$= \sum_x d_{(m)}^{\pi}(x) \sum_a \frac{\partial \pi_\Omega^{(m)}(x, a)}{\partial \Omega} Q_{(m)}^{\pi_\Omega}(x, a)$$

as derived in Sutton et al. (2000) for MDP model $m$. Now if we substitute this result into Equation 17, we get $\frac{\partial \rho(\pi_\Omega)}{\partial \Omega}$ defined in Equation 18.

$$\frac{\partial \rho(\pi_\Omega)}{\partial \Omega} = \sum_{m \in M} \mu(m) \frac{\partial}{\partial \Omega} J^{(m)}(\pi_\Omega)$$

$$= \sum_{m \in M} \mu(m) \sum_x d_{(m)}^{\pi}(x) \sum_a \frac{\partial \pi_\Omega^{(m)}(x, a)}{\partial \Omega} Q_{(m)}^{\pi_\Omega}(x, a) \tag{18}$$

Using this result, we can now incorporate function approximation as shown in Sutton et al. (2000) where we first state that when a process has converged to a local optimum, we have the following result:

$$\sum_{m \in M} \mu(m) \sum_x d_{(m)}^{\pi}(x) \sum_a \frac{\partial \pi_\Omega^{(m)}(x, a)}{\partial \Omega} \tag{19}$$

$$\left[ Q_{(m)}^{\pi_\Omega}(x, a) - J_w^{(m)}(x, a) \right] \frac{\partial J_w^{(m)}(x, a)}{\partial w} = 0$$

where $J_w(x, a)$ is the approximation to $Q^{\pi_\Omega}$. It then follows that provided $J_w^{(m)}(x, a)$ obeys the compatibility condition, we can easily incorporate function approximation into the gradient of the objective function and we get:

$$\frac{\partial \rho(\pi_\Omega)}{\partial \Omega} = \sum_{m \in M} \mu(m) \sum_x d_{(m)}^{\pi}(x) \sum_a \frac{\partial \pi_\Omega^{(m)}(x, a)}{\partial \Omega} J_w^{(m)}(x, a)$$

Using this formula, we can converge to a local minimum based on the following conditions. The step-size $\alpha_k$ has two requirements: (1) $\lim_{k \to \infty} \alpha_k = 0$ and (2) $\sum_k \alpha_k = \infty$. The second derivative of the policy is bounded and we have bounded rewards. Then, the sequence $\{\rho(\pi_{\Omega,k})\}_{k=0}^{\infty}$ converges such that the limit $\lim_{k \to \infty} \frac{\partial \rho(\pi_{\Omega,k})}{\partial \Omega} = 0$. This is proven by proposition 3.6 in Bertsekas (1996). The following update rule results:

$$\Omega_{k+1} = \Omega_k + \alpha_k \frac{\partial \rho(\pi_{\Omega,k})}{\partial \Omega}$$

$$= \Omega_k + \alpha_k \sum_{m \in M} \mu(m) \sum_x d_{(m)}^{\pi}(x) \sum_a \frac{\partial \pi_{\Omega}^{(m)}(x,a)}{\partial \Omega} J_{w_k}^{(m)}(s,a)$$

where $w_k$ is obtained from:

$$w_k = w \ ,$$

such that,

$$\sum_{m \in M} \mu(m) \sum_x d_{(m)}^{\pi}(x) \sum_a \frac{\partial \pi_{\Omega}^{(m)}(x,a)}{\partial \Omega}$$

$$\left[ Q_{(m)}^{\pi_{\Omega}}(x,a) - J_w^{(m)}(x,a) \right] \frac{\partial J_w^{(m)}(x,a)}{\partial w} = 0$$

$\square$

## 6   Experiments

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

The single fourier feature is $\psi_{x,m} = [\sin(\pi x^T a)]$ where $a = [3, 0]$, the temperature parameter for the intra-skill policy parameters $\alpha_\theta = 1$; for the hyperplane parameters, the temperature $\alpha_\beta = 20$. The gradient update stepsize for the hyperplane is 1 and intra-skill parameters update stepsize is 10.

**RoboCup Domain:** The RoboCup 2D soccer simulation domain (Akiyama & Nakashima, 2014) is a 2D soccer field (Figure 1$e$) with two opposing teams. We utilized three RoboCup sub-domains [3] R1, R2 and R3 as mentioned previously. In these sub-domains, a striker (the agent) needs to learn to dribble the ball and try and score goals past the goalkeeper. **State space:** R1 domain - the continuous locations of the striker $\langle x_{striker}, y_{striker} \rangle$ , the ball $\langle x_{ball}, y_{ball} \rangle$, the goalkeeper $\langle x_{goalkeeper}, y_{goalkeeper} \rangle$ and the constant goal location $\langle x_{goal}, y_{goal} \rangle$. R2 domain - we have the addition of the defender's location $\langle x_{defender}, y_{defender} \rangle$ to the state space. R3 domain - we add the locations of two defenders. **Features:** For the R1 domain, we tested both a linear and degree two polynomial feature representation for the hyperplanes. For the R2 and R3 domains, we also utilized a degree two polynomial hyperplane feature representation. **Actions:** The striker has three actions which are (1) move to the ball (**M**), (2) move to the ball and dribble towards the goal (**D**) (3) move to the ball and shoot towards the goal (**S**). **Rewards:** The reward setup is consistent with logical football strategies (Hausknecht & Stone, 2015; Bai et al., 2012). Small negative (positive) rewards for shooting from outside (inside) the box and dribbling when inside (outside) the box. Large negative rewards for losing possession and kicking the ball out of bounds. Large positive reward for scoring.

For the R1, R2 and R3 RoboCup domains, the temperature parameter for the intra-skill policy parameters $\alpha_\theta = 2$; for the hyperplane parameters, the temperature $\alpha_\beta = 100$. The gradient update stepsize for the hyperplane is 40 and intra-skill parameters update stepsize is 0.5.

**Learning Offline**: RoboCup's simulator is not well suited to learning online since actions can only be broadcast to the players and feedback received from the game every 30 ms (with various speedups). This is problematic if you wish to learn on tens of thousands of episodes. We therefore wanted to test the ability of ASAP to learn using *offline* trajectories. These trajectories are generated by an approximately optimal hand-coded scoring controller which is provided to the striker. Using this

Figure 4: The RoboCup R1 Domain: The average reward for the R1 domain compared to initial misspecified model and the approximately optimal controller for polynomial hyperplane features.

Figure 5: The average goal scoring ratio for $1000$ trials for ($a$) The RoboCup R1 domain, ($b$) The R2 domain and, ($c$) The R3 domain. In each figure **I** refers to the goal scoring ratio for fixed, initial misspecified skills and SPs; **A** refers to ASAP's performance after learning near-optimal skills and SPs from the misspecified model; **O** is the performance of the approximately optimal controller.

speedup, over $100,000$ trajectories can be gathered in an hour. However, it is well known that policy gradient algorithms struggle with offline learning. ASAP managed to learn near-optimal SPs and skill sets for both the R1, R2 and R3 domains, using offline trajectories, as seen in Figure $3f$ and Figure 5 respectively. These results were consistently attained over five datasets. In each case, the agent learned that it should **D: Dribble** in the yellow SP and should **S: Shoot** in the semi circular SP near the goal.

**Different SP Optimas**: Since ASAP attains a locally optimal solution, it may sometimes learn different SPs. For the polynomial hyperplane feature representation, ASAP attained two different solutions as shown in Figure $3g(1)$ as well as Figure $3g(2)$, respectively. Both achieve near optimal performance compared to the approximately optimal scoring controller (see supplementary material). For the linear feature representation, the SPs and skill set in Figure $3g(3)$ is obtained and achieved near-optimal performance (Figure $3j$), outperforming the polynomial representation.

The average ratio of goals scored over $1000$ episodes is $79\%$ for the learned ASAP policy compared to $91\%$ for the approximately optimal scoring controller and $18\%$ for ASAP evaluated on the initial misspecified model. **SP Sensitivity**: In the R2 domain, an additional player (the defender) is added to the game. It is expected that the presence of the defender will affect the shape of the learned SPs. ASAP again learns intuitive SPs. However, the shape of the learned SPs change based on the pre-defined hyperplane feature vector $\psi_{m,x}$. Figure $3h(1)$ shows the learned SPs when the location of the defender is *not* used as a hyperplane feature. When the $x$ location of the defender is utilized, the 'flatter' SPs are learned in Figure $3h(2)$. Using the $y$ location of the defender as a hyperplane feature causes the hyperplane offset shown in Figure $3h(3)$. This is due to the striker learning to dribble around the defender in order to score a goal as seen in Figure $3i$. Finally, taking the $\langle x, y \rangle$ location of the defender into account results in the 'squashed' SPs shown in Figure $3h(4)$ clearly showing the sensitivity and adaptability of ASAP to dynamic factors in the environment.

# 7 Discussion

We have presented the Adaptive Skills, Adaptive Partitions (ASAP) framework that is able to automatically compose skills together and learns a near-optimal skill set and skill partitions (the inter-skill policy) simultaneously to correct an initially misspecified model. We derived the gradient update rules for both skill and skill hyperplane parameters and incorporated them into a policy gradient framework. This is possible due to our definition of a generalized trajectory. In addition, ASAP has shown the potential to learn across multiple tasks as well as automatically reuse skills. These are the necessary requirements for a truly general skill learning framework and can be applied to lifelong learning problems (Ammar et al., 2015; Thrun & Mitchell, 1995). An exciting extension of this work is to incorporate it into a Deep Reinforcement Learning framework, where both the skills and ASAP policy can be represented as deep networks.

### Acknowledgements

The research leading to these results has received funding from the European Research Council under the European Union's Seventh Framework Program (FP/2007-2013) / ERC Grant Agreement n. 306638.

## Footnotes

[1]These expectations can easily be sampled (see supplementary material).

[2]AC-PG works well in practice and can be trivially incorporated into ASAP with convergence guarantees

[3]https://github.com/mhauskn/HFO.git