[Reviews · NeurIPS 2016]

Reviewer 1

Summary

This paper addresses the problem of learning temporally extended actions ("skills") as well as when to apply them in reinforcement learning. The main idea is to smoothly parameterize every component of the skill architecture (the state partition, the skill policies, and the assignment of partitions to skills) and to directly adapt those parameters using policy gradient. Experimental results show that the ASAP algorithm can effectively adapt its arbitrary initial skills to suit its environment.

Qualitative Assessment

***After Author Response*** I would only add that I hope the authors will take pains to clarify issues surrounding the number of partitions and policies if this paper is accepted. Several of the reviewers brought this up, and since the reviewers are at least somewhat representative of the paper's audience, it should be carefully addressed in the paper. I understand the authors to be saying that the algorithm is capable of making some partitions vanishingly small, and is capable of making some options vanishingly unlikely to be executed. In that sense, yes, ASAP can effectively learn how many partitions/policies are needed. However, I think the authors are being a little bit disingenuous about this point. For instance, ASAP can't create *more* partitions if they turn out to be necessary, which is I think what people mean when they say something like "how do you pick the correct number?". Also, even if ASAP might eventually find a structure that effectively uses only a few policies, on the way to get there (and actually even once it is there) it has to maintain and adapt the parameters for exponentially many policies. So that would suggest that it is not safe to just throw a whole bunch of partitions at it and let it sort it out. It seems that one does need to carefully consider how many partitions to start with -- not too few, not too many. So one of two things is happening. If, as I suspect, this is a genuine limitation of this approach, it's important that the authors straightforwardly and clearly acknowledge it as a limitation; the results are interesting anyway, and it points the way to remaining open questions. If basically all of the reviewers have misunderstood something and this is not a practical limitation, then there must be a clarity issue in the paper which must be addressed. ---Technical Quality--- As far as I can tell the paper is technically sound. The derivations of the gradients are wholly contained within the supplemental material, and I did not pore over every detail, but they seemed correct and reasonable. The experiments are well-constructed and do effectively support the main claims. I will say that the claim that ASAP re-uses skills seemed a bit weaker than the others. In "Three Rooms" the system apparently learned to apply the same skill in two different partitions, which is cool, but in the "Flipped Two Rooms" domain, the system apparently kept the hyperplane while re-learning the same two skills (just in the opposite regions), which is less cool. I think it's fair to say that the experiments demonstrate that skills *can* be re-used, but they also seem to show that this is highly dependent on the parameterization and search procedure. In other words, this framework permits skill re-use, but does nothing in particular to encourage/facilitate it. ---Novelty--- The most similar work I am aware of is from Bacon and Precup (which the paper cites). They similarly parameterize the option policies, option termination conditions, and overall policy so all components can be adapted via policy gradient. This paper deals directly with continuous state spaces (defining state partitions using hyperplanes), which Bacon and Precup did not. This would not seem like sufficient novelty on its own, however the Bacon and Precup paper appeared in a workshop proceedings, not in an archival publication. Other than Bacon & Precup, as far as I am aware this approach of directly the entire skill architecture via PG is novel (and interesting). ---Impact--- I believe this paper has the potential for substantial impact. Certainly the problems of learning and applying skills is an important one. This paper's approach of folding all of the relevant architectural decisions into one learning process is intriguing and shows demonstrated promise. This particular algorithm definitely has weaknesses; the exponental number of skill policies to learn is probably chief among them. However, the formulation of skill learning as purely a highly structured parameterization of the policy is a core idea that I can easily see being further explored in follow-up work. Furthermore, the experiments show some emergent desirable properties of this relatively straightforward approach (e.g. skill re-use, learning that fewer partitions are necessary) that immediately suggest follow-up research into how to more systematically encourage these behaviors. ---Clarity--- I found the paper to be very well written. I thought the approach was well-motivated and clearly described, as were the presented results. I felt that leaving the derivations for the supplementary appendix was sensible, since these are mostly algebra, and the derivation techniques applied are not really critical to the contribution. Even though some of the experimental results are left to the appendix, I felt that the authors did a good job of describing the results, and selected appropriate figures that could be left out without too much harm to the reader's flow. I will say that I personally don't like the format of making the supplementary materials simply a longer version of the paper. As a reader, it makes it harder to flip to the appendix to see what I want to see. Also, it is confusing that the supplementary version still refers to the supplementary materials! A minor question: Why is Masson & Konidaris (2015) in Table 1? It is nowhere discussed in the paper, and, while I can see the relevance, it does not seem to be directly addressing the issue of skill learning in the same way as the others. ---Summary of Revew--- This is a novel approach to skill learning that is intruiging in its directness and that shows significant promise. While there are some drawbacks to the specific algorithm presented here, the core idea is flexible and seems likely to inspire further investigation.

Confidence in this Review

3-Expert (read the paper in detail, know the area, quite certain of my opinion)


Reviewer 2

Summary

In this paper the authors present their Adaptive Skills, Adaptive Partition (ASAP) framework. ASAP is a policy gradient learning framework for learning simultaneously to compose skills (options, temporally extended actions) and the inter-skill policy to correct initially misspecified models. This is accomplish by partitioning the state space using hyperplanes that divide the state space into skill partitions in which the skills are used. The authors define the gradient update rules for skill and hyperplane parameters. This is made possible, by defining generalized trajectories that also contain the skill executed. The framework was evaluated on multiple toy problems. The authors were able to show that in a very simple setup, the algorithm is able to learn near-optimal policies, that previous learned policies decrease the number of episodes to learn similar problems, and that the framework is able to identify re-usable skills.

Qualitative Assessment

* One of the biggest concerns of the algorithm is the scalability to real world applications. The presented test task were all very simple environments with small state-action spaces. The skills and inter-skill policy that needed to be learned were also kept very simple. The only example that had potential to show real world abilities was the robot soccer task. Here however, the action were skills itself such that it was again kept very simple. * Another concern is that many parameters need to be chosen by the designer and that the performance is critically defined by those parameters: the number and features of the hyperplanes. * I don't get the point with the distribution of the multiple tasks. Should that not be defined very clearly by the task at hand? *Line 83: A should be lower case like on the page before *Line 97 to 101: But that also means that the state space must allow for such a separation. *Line 107-108: "Note that this is .." -> I would say that is one of the most important points that need to be addressed. The ability to partition the space and allow to define a probability distribution for the skills based on the state space is crucial to the success in real world applications. *Line 147: More explanation about the necessity of z is necessary. *Line 165-166: "This is a slight ..." The authors might want to comment a little bit more on this. *Experiments: How was the misspecified partitioning/model defined for all the experiments? *Experiments: What is the number of episodes to learn the optimal policy for all the experiments? *Experiments: How was the optimal policy derived? *Experiments: How in general is the designer suppose to come up with good hyperplane features? *Experiments: Reward for more complex tasks critically and needs to be well designed. Scalability questionable.

Confidence in this Review

2-Confident (read it all; understood it all reasonably well)


Reviewer 3

Summary

This is a reinforcement learning paper. The aim is to learn options and how to combine them to solve a task. Results of experiments are provided.

Qualitative Assessment

This paper considers a very important problem, aiming towards solving "complex" reinforcement tasks that require the combination of a set of options (skills). This idea is to split the domain into areas in which an option has to be learnt, and connect the options between neighboring domains. In itself, this idea is common sense, but I like that people work on that sort of ideas. I am less satisfied with the fact that areas are separated by hyperplanes, and that the number of hyperplanes is set and not learned. Having 2^K skills to learn is redibitory: the authors write that far fewer skills need to be learned in practice, but this is hand-coded, not learned: how can we deal with problems that are not toy problems for which the human being provides a correct and very small number of hyperplanes to use? The experimental part is rather weak, the robocup being more significant, while the other domains illustrate how ASAP works. I wonder how a Q-learning would behave in such small domains. We have seen robots learning by reinforcement to play soccer for real (Riedmiller, ...): they don't need ASAP for that. What would ASAP bring to these works?

Confidence in this Review

2-Confident (read it all; understood it all reasonably well)


Reviewer 4

Summary

A new learning framework called ASAP was introduced. ASAP enables learning skills and and inter-skill policies, given an initially misspecified model. Learning is performed using policy gradient method. The proposed algorithm was verified on a number of tasks and has shown ability to learn across multiple tasks as well as automatically reuse skills.

Qualitative Assessment

This is solid paper with a good theoretical contribution as well as promising experimental results. As most of the derivations depend on the choice of the Bernoulli distribution for the probability of executing a skill and the Gibbs distribution for the intra-skill policy, it would be worthwhile mentioning what motivated this choice and what other possibilities could have been used instead.

Confidence in this Review

1-Less confident (might not have understood significant parts)


Reviewer 5

Summary

The authors present in this work a reinforcement learning (RL) framework for continuous state space where skills as well as partitioning of the continuous state space are obtained in an adaptive fashion. They declare five properties that a truly skill learning framework must have and comparing with the state of the art they claim their algorithm (ASAP) bears them. They explicitly state that the main contributions of this paper are ASAP 1) automatically corrects a misspecified model, learns a set of near-optimal skills, automatically composes skills together, 2) achieves multi-task task learning in continuous state spaces, 3) learns where to reuse skills and provides theoretical convergence guarantees.

Qualitative Assessment

Using the background theory of the neural networks, the authors extend the general hierarchical reinforcement learning framework for continuous state spaces to handle non-stationary environments and making use of transfer learning. While making this nice extension, they do not mention about the hierarchical reinforcement learning (HRL) concept. They cite rightfully the 1999 paper of Sutton et.al., use concepts such as "temporally extended actions," "temporal abstraction" as expected; but they skip the concept of HRL. This way they may miss several references of some relevance. One reference that deserves citation in this work is "Hierarchical Reinforcement Learning with Context Detection (HRL-CD)" of Yucesoy et.al, (Yücesoy, Yigit E; Tümer, M Borahan. International Journal of Machine Learning and Computing v:5 n:5 (Oct 2015): 353-358) which presents a model-based agent that autonomously learns options/skills in a non-stationary environment with a discrete state space. Comments/suggestions: In the proof of Theorem 1: At the fourth line of equations the authors use the log likelihood trick defined in the 1968 paper of Aleksandrov, Syosev and Shemeneva. The authors should mention this in the supplementary material in the proof. Why are the second and third to the last lines are identical at the end of the proof?

Confidence in this Review

3-Expert (read the paper in detail, know the area, quite certain of my opinion)


Reviewer 6

Summary

This paper proposes a kind of modular reinforcement learning that can learn a set of lower-level policies and an upper-level module selection policy at the same time. The authors propose the ASAP policy that consists of two components. One is the probability of selecting a skill and the other is a stochastic policy. Once the combined policy is formulated, the standard policy gradient method is applied in order to derive the learning algorithm. The proposed method is evaluated by some toy problems and the RoboCup domain. Although the proposed method is not compared with other methods, it can learn skills and skill selection at the same time.

Qualitative Assessment

Major comments: The contribution of the paper is to derive the ASAP policy (4) where P_\beta (i|x, m) is interpreted as a mixing weight to combine multiple policies. I think this is a kind of modular reinforcement learning and there are several methods: Thomas, P.S. and Barto, A.G. (2012). Motor primitive discovery. In Proc. of IEEE International Conference on Development and Learning and Epigenetic Robotics. Their method optimizes multiple skills and mixing weights to combine the multiple skills by a policy gradient method. Although there are some differences between Thomas’s method and the ASAP method, it is better to mention it. The theoretical part is well written, but I do not fully understand how the algorithm is implemented. Algorithm 1 in page 6 is based on the policy gradient method while the first paragraph in Section 6 claims that ASAP is implemented by Actor-Critic Policy Gradient. If the authors use Actor-Critic Policy Gradient, it is better to show the algorithm in the main manuscript. In addition, the state-action value function should be estimated in the case of actor-critic policy gradient. How did you estimate the value function? Theorem 2 of Sutton et al. (2000) shows the function approximator to the value function is compatible with the policy parameterization. Do you show the similar theorem in the proposed method? Experiments: The proposed method should be compared with other methods. In particular, I’m interested in the difference between the proposed method and the plain approach. For example, the 2R task can be solved by a normal policy gradient method. Radial basis functions are uniformly distributed in the two dimensional state space and the stochastic policy (3) can be constructed in the standard way. I could not find the basis functions \phi_{x,a} to represent intra-skill policies although \psi_{x, m} was explained. Please specify them. Figure 1(h) shows the example that the ASAP method can reuse skills. This result seems interesting, but it severely depends on the representation ability of SPs. In this example, the Fourier features are used to construct the partition shown in Figure 1(h). Is it easy to prepare such features without prior knowledge? Minor comments: Eq.(1): p(i|x, m) -> P_\beta (i|x, m) Eq.(4): p_\beta (i|x, m) -> P_\beta (i|x, m)

Confidence in this Review

3-Expert (read the paper in detail, know the area, quite certain of my opinion)